# Evaluating Classification Systems of Diabetic Foot Ulcer Severity: A 12-Year Retrospective Study on Factors Impacting Survival

**DOI:** 10.3390/healthcare11142077

**Published:** 2023-07-20

**Authors:** Otilia Niță, Lidia Iuliana Arhire, Laura Mihalache, Alina Delia Popa, George Niță, Andreea Gherasim, Mariana Graur

**Affiliations:** 1Faculty of Medicine, University of Medicine and Pharmacy “Grigore T Popa”, 700115 Iasi, Romania; otilia.nita@umfiasi.ro (O.N.); lidia.graur@umfiasi.ro (L.I.A.); andreea.gherasim@umfiasi.ro (G.N.); george.nita@umfiasi.ro (A.G.); 2Faculty of Medicine and Biological Sciences, University “Ștefan cel Mare” of Suceava, 720229 Suceava, Romania; graur.mariana@gmail.com

**Keywords:** diabetic foot ulcers, survival, University of Texas Staging System, Wagner–Meggitt classification, SINBAD classification system and score, mortality

## Abstract

(1) Background: This study examines the survival of patients after their first presentation with diabetic foot ulcers (DFUs) to the regional Diabetes, Nutrition, and Metabolic Diseases Clinic within the Emergency Clinical Hospital “Sf. Spiridon”, Iaşi, and analyzes the factors associated with this outcome. (2) Methods: In this retrospective study, patients with DFUs consecutively referred between 1 January 2007 and 31 December 2017 were followed up until 31 December 2020 (for 13 years). The study group included 659 subjects. (3) Results: During the study period, there were 278 deaths (42.2%) and the average survival time was 9 years. The length of hospitalization, diabetic nephropathy, chronic kidney disease, glomerular filtration rate, cardiovascular disease, hypertension, anemia, and DFU severity were the most significant contributors to the increase in mortality. Patients with severe ulcers, meaning DFUs involving the tendon, joint, or bone, had a higher mortality risk than those with superficial or pre-ulcerative lesions on initial presentation (Texas classification HR = 1.963, 95% CI: 1.063–3.617; Wagner–Meggitt classification HR = 1.889, 95% CI: 1.024–3.417, SINBAD Classification System and Score HR = 2.333, 95% CI: 1.258–4.326) after adjusting for confounding factors. (4) Conclusions: The findings of this study suggested that patients presenting with severe ulcers involving the tendon, joint, or bone exhibited a significantly higher risk of mortality, even when potential confounders were taken into consideration.

## 1. Introduction

The prevalence of diabetes is increasing worldwide; therefore, one of the main goals of the World Health Organization (WHO) is fighting against this disease due to the impressive data related to diabetic foot epidemiology. Diabetes is the primary cause of 50 to 70% of all lower-limb amputations; 15% of diabetic patients will acquire a foot ulcer and 25–50% of them will need an amputation as a result of complications [1]. One amputation is performed every thirty seconds on a person with diabetes somewhere in the world, according to estimates [2]. Diabetic foot disease is a complex, specific complication in which neuropathy, peripheral artery disease, immunopathy, and infection [1,2] have the potential to injure all anatomical components of the lower limb: skin, nails, adipose tissue, muscles, fascia, vessels, nerves, bones, and joints [3,4]. Poor glycemic control, cigarette smoking, impaired vision, peripheral neuropathy with loss of protective sensation, peripheral artery disease, foot deformations, previous foot ulcers, and history of amputation are the most significant risk factors for diabetic foot ulcers (DFUs) [5,6].

The prevalence of foot ulcers varies widely, from 1% in Europe and North America to 11% in certain African countries [7]. According to estimates, 3.60% of diabetic patients in Romania have had an amputation, and 14.85% of them have a history of foot ulcers [8]. The risk of recurrence of foot ulcers is a subject of considerable interest due to the high cost of treatment, which exceeds the expenses associated with managing many common cancers [9]. The costs are related to the economic burden imposed on the patient, the patient’s family, and the healthcare system [10]. According to the International Diabetes Federation, one-third of the costs associated with diabetes are attributable to foot ulcers. When compared with diabetes patients without foot ulcers, the costs are 5.4 times higher [1]. The risk for ulcer recurrence is considerable: roughly 40% of patients develop a new ulcer during the first year, and nearly 65% experience a new lesion within five years [9]. Previous foot ulcers and their location, peripheral artery disease, loss of protective sensation and vibration perception, presence of osteomyelitis, depression, elevated C-reactive protein (CRP), and uncontrolled glycated hemoglobin (HbA1c) are risk factors independently associated with ulcer recurrence [8].

A recent meta-analysis established that the risk factors associated with 5-year mortality in patients with DFU were age, peripheral artery disease, chronic kidney disease, end-stage renal disease, amputation, and history of cardiovascular disease [11].

Clinical classification systems for diabetic foot (the University of Texas Staging System for Diabetic Foot Ulcers; Perfusion, Extent, Depth, Infection, and Sensation (PEDIS) Classification System and Score; IWGDF Guidelines on the prevention and management of diabetes-related foot disease; SINBAD (Site, Ischaemia, Bacterial infection, Area, and Depth) classification) [12,13] include risk factors for recurrence, such as the area of the wound’s surface, its perfusion, and its depth. The Wagner–Meggitt classification is the most common classification system used to predict amputation, according to a recent systematic review of DFU classification, but there are still insufficient studies attempting to correlate DFU severity with ulcer-free survival, hospitalization, and mortality [14]. The available evidence remains insufficient in providing a comprehensive understanding of the relationship between DFU classification systems and these important clinical outcomes. Therefore, further research is warranted to bridge this knowledge gap and enhance our ability to predict mortality in patients with DFU accurately [14,15,16]. 

Our study’s primary objective was to evaluate the clinical and biological factors that may influence the survival outcomes of individuals following the onset of diabetic foot ulcers. This study focused on estimating survival rates and identifying mortality predictors among a cohort of patients hospitalized consecutively over a 10-year period at a single regional center specialized in the treatment of DFUs. We used five classification systems to assess the severity of DFU, developing mortality prediction models for each of them. By evaluating these variables, a thorough understanding of the prognosis and influential factors associated with diabetic foot ulcers in hospitalized patients with diabetes may be attained.

## 2. Materials and Methods

We conducted a study on patients with DFUs who were consecutively referred to the Diabetes, Nutrition, and Metabolic Diseases Clinic within the Emergency Clinical Hospital “Sf. Spiridon”, Iași, between 1 January 2007 and 31 December 2017. Patients admitted to our regional center were from 8 of the 41 counties in the north-eastern region of Romania. Patients transferred from other clinics or referred through the emergency department or by their diabetologist or family doctor were followed up until death or the final date recorded in the electronic case system for a period of three more years (until 31 December 2020). If there were multiple admissions during the study period, the first hospitalization was considered for the analysis.

### 2.1. Data Collection

The inclusion criteria for data collection were as follows: -Patients over 18 years of age, diagnosed with any type of diabetes (type 1, type 2, or other specific types);-Patients admitted to the Diabetes, Nutrition, and Metabolic Diseases Clinic during the aforementioned time period; -Patients with an International Statistical Classification of Diseases, 10th Revision (ICD-10) [17] code corresponding to diabetic foot pathology in the discharge diagnostic, or soft tissue lesions. 

The exclusion criteria were as follows: previous foot ulcers or amputation, active neoplasia, end-stage organ diseases (end-stage respiratory disease, hepatic failure, end-stage renal disease, and heart failure III-IV NYHA), dementia or psychiatric illnesses, and collagen vascular diseases.

A database including all patients whose discharge files contained ICD-10 codes that could have been linked to diabetic foot pathology (lower-limb cellulitis, ulceration, or gangrene, peripheral artery disease, peripheral neuropathy, etc.) was developed with the assistance of the hospital’s statistics service. A discharge diagnosis analysis was performed, and patients without a diagnosis related to DFU were removed from the database. Then, hospitalizations with the same personal identification code were analyzed, and the first hospitalization was chosen, with the others excluded from the main analysis, but considered as the number of readmissions for the same patient, representing a variable related to the progression of diabetic foot. Thus, 659 patients were entered into the database (Figure 1).

We included in our analyses the following potential risk factors associated with DFU outcome:Social and demographic factors: area of residence (rural or urban), sex, and age;History of diabetes: disease duration, type of treatment (metformin, non-insulin antidiabetic agents, insulin, and association of insulin with other antidiabetic agents);Presence of specific diabetic chronic complications (retinopathy, nephropathy, neuropathy, peripheral artery disease), high blood pressure (HBP), and cardiovascular disease evaluated during the first hospitalization;Biological markers: glycated hemoglobin (HbA1c), inflammation markers (white cell blood count, fibrinogen, CRP, ferritin), lipid profile, renal function, total proteins, albumin, ASAT, ALAT, and uric acid;Clinical characteristics of wounds: location (toes, under metatarsals, dorsum of foot, heel), depth (using a sterile blunt nasal probe), extension (evaluating the surface of the wound), and perfusion;Infection: the presence of symptoms and signs of inflammation and/or pus, abnormal specific biochemical test results, and the presence of osteolysis on foot X-ray.In this study, five clinical classification systems for diabetic foot were used:
a.The University of Texas Staging System for Diabetic Foot Ulcers uses four grades (0—pre-ulcerative or post-ulcerative wound entirely epithelialized; 1—superficial lesion, not involving the tendon, capsule, or bone; 2—wound penetrating tendon or capsule; and 3—wound penetrating bone or joint), which are influenced by the existence of infection (Stage B), ischemia (Stage C), or both (Stage D) [18]. Scores higher than BII are considered predictors of foot amputation [19];b.The Wagner–Meggitt classification of foot ulcers evaluates the clinical characteristics of foot ulcers (extent and depth) [20]; c.The Saint Elian Wound Score System (SEWSS) [21] includes 10 variables organized into three categories: anatomical factors, aggravating factors, and factors related to ulceration. The first category evaluates anatomical factors, the site, and topographical aspects. In the second category of aggravating factors, four variables are included: ischemia, infection, edema, and neuropathy. In the case of infection, the indications of the IDSA (Infectious Diseases Society of America) [12] are used; therefore, a mild infection is one that affects only the superficial layers of the skin and is characterized by erythema between 0.5 mm and 2 cm, induration, pain, local heat, and purulent secretions [21]. Moderate infection is identified by erythema > 2 cm, and infection of the muscle, tendons, bone, or joint. Osteomyelitis is diagnosed via radiography or biopsy. Severe infection is characterized by a systemic inflammatory response or severe metabolic disorders that require hospitalization or are difficult to manage [21]. The third category of ulceration-related factors includes three variables: depth, area, and healing phase. The result of this scoring system is the sum of all criteria met by ulceration, ranging from six to thirty points. The severity of the ulceration is established by the obtained value, a distinct prognosis being proposed for each of the three degrees: I—mild, probable successful cure; II—moderate, with a partial threat to the foot, where the prognosis depends on the “state-of-the-art” therapies used and the patient’s biological response; and III—severe and life-threatening, where the prognosis does not depend on “state-of-the-art” therapies as a result of the patient’s poor biological response [22];d.The SINBAD Classification System and Score is an acronym that comes from six elements graded according to their severity: ulcer Site, Ischemia, Neuropathy, Bacterial infection, Area, and Depth [23]. The total score ranges between 0 and 6, which is divided into three categories related to the risk of lower limb amputation: low grade, 0–2; moderate grade, 3–4; and high grade, 5–6 [24,25];e.The Society for Vascular Surgery Lower Extremity Threatened Limb Classification System (WIfI) is a grading system that evaluates the severity of a wound (ranging from 0 for no ulcer to 3 for an extensive, deep ulcer or gangrene involving the forefoot and/or midfoot), ischemia (ranging from 0 to 3, evaluated through the ankle–brachial index, ankle systolic pressure, or transcutaneous oximetry), and foot infection (from absent to severe: limb- and/or life-threatening) [26]. Based on the associations of the scores obtained for each grade, the WIfI stage (1–5) is then determined [27];The necessity of amputation of the lower limb for DFU during the first hospitalization was registered. Patients who needed a limb amputation were transferred to a surgical clinic in the same hospital.

The intended outcome, death or survival, was recorded in the database, being extracted from the regional registry of diabetic patients, so that the authors had access to all information regarding their subsequent evolution.

### 2.2. Statistical Analyses

Data analysis was performed using the Statistical Package for Social Sciences (SPSS software version 20) [28]. In the survival analysis, the Hosmer–Lemeshow test was used to assess the strength of the models created. Kaplan–Meier and log-rank tests were used to compare each group determined by the presence or absence of a presumed risk factor on survival time, i.e., the time until death from any cause. Univariate and multivariate Cox regressions were used to determine the hazard risk ratio (HR) and 95% confidence interval (CI) in the three models, which included the following as predictors for the survival endpoint: the DFU’s severity, sex, diabetes duration, age, albumin, glomerular filtration rate, hemoglobin, presence of hypertension, and cardiovascular disease. Time to death was measured as the number of months from inclusion to death from all causes. For all analyses, a *p*-value of < 0.05 was considered significant. An HR value > 1 corresponds to an increase in relative risk compared with a reference subject.

### 2.3. Ethical Considerations

This study was approved by the Ethics Commission of the University of Medicine and Pharmacy “Gr. T. Popa”, Iaşi, approval number 2324/19 January 2017. Informed consent from patients was not requested because it was a retrospective, observational study. The identification data of the participants have not been and will not be used for any other purpose.

## 3. Results

This study was conducted on 659 subjects with a mean age of 61.34 ± 11.06 years and the majority of participants were men (66%). Most patients (51.4%) came from urban areas. Among the patients, a high proportion had had type 2 diabetes (86.19%), followed by type 1 diabetes (12.4%) and other specific types of diabetes (1.67%). Patients with type 1 diabetes had a statistically significantly longer duration of the disease (17.59 years) compared with people with type 2 diabetes (10.37 years) and other specific types of diabetes (10.09 years) (*p* < 0.05) (Table 1). A percentage of 77.5% had a HbA1c > 7%, reflecting inadequate glycemic control. In terms of microangiopathic complications, 65.4% had a diagnosis of diabetic retinopathy, 31.9% had a diagnosis of diabetic nephropathy, 35.1% had a diagnosis of chronic kidney disease, and 63.3% had a diagnosis of peripheral artery disease (27.3% stage IV). Sensorimotor peripheral polyneuropathy was diagnosed in 93.9% of patients (61.3% male and 32.6% female), with no statistically significant differences between sexes (*p* = 0.113). Charcot’s foot had a low incidence in our study group (3.2%). Cardiovascular disease was identified in 59.8% of participants. Most patients included in our study had high blood pressure (73.3%). Obesity was identified in 31.6% of the total group. 

Regarding the type of hospitalization, we noticed that 73.6% were emergency admissions, a smaller percentage (0.61%) were hospitalized via transfer from other departments, and the rest were referred to our clinic by family doctors or other specialists. Diabetic ketoacidosis on admission was present in 14.9% of participants, with a higher frequency in men (9.5%). The average length of hospitalization was 19.68 ± 13.38 days. In the studied group, some cases had subsequent readmissions (a percentage of 29.89%). 

In addition to the usual discharge, which was the majority of cases (89.53%), 3.64% were transferred to other departments, 1.37% died during the first hospitalization, and 4.86% were discharged on request.

According to the University of Texas Staging System for Diabetic Foot Ulcers, 0.2% of DFUs were classified as AIII and the same percentage (0.2%) were categorized as B0. The majority of patients (21.2%) had a stage BI ulcer (Table 2). There were no statistically significant differences between sexes (*p* = 0.418).

According to the Wagner–Meggitt classification, 32.4% of patients had grade 1 ulcers, and 29.2% had grade 2 ulcers. The smallest number of cases had grade 0 ulcers (Table 2). There were no statistically significant differences between sexes (*p* = 0.940).

The SEWSS classification (mild, moderate, and severe) evaluation revealed that 85.7% of all cases were moderate., with no differences between men and women (*p* = 0.625) (Table 2).

According to the SINBAD Classification System and Score, only 8.2% of the patients had a high-grade DFU. Almost similar percentages were observed for low- and moderate-grade ulcers (43.6% and 40.8%, respectively).

When the WIfI Classification system was applied, we identified that 29.1% of the patients had stage 4 ulcers

In our sample of 659 subjects, 278 deaths were recorded during the 13-year period (42.2%). The average survival time was 9 years, with a median of 12 years (Table 3, Figure 2).

When analyzing the general characteristics of the studied group, statistically significant differences in survival were noticed for the age categories (HR = 1.046, *p* < 0.001) (Table 4). Patients with renal impairment (diabetic nephropathy: HR = 1.36, *p* = 0.12 or chronic kidney disease) and cardiovascular diseases (HR = 1.75, *p* < 0.001) had a shorter average survival time (Table 4, Figure 3).

Patients with Gram-positive aerobic bacteria wound infection had a longer survival time (*p* = 0.044). Participants with the Texas A0 to BI stages had a statistically significantly longer survival time than those with stages BII through DIII. There were no differences in survival time between grades 3, 4, and 5, the first two grades in the Wagner–Meggitt classification. The Saint Elian Wound Score System was associated with higher mortality risk (Figure 3). A high grade in the SINBAD Classification System or a high risk (stage 4–5) evaluated with the WIfI Classification was related to an increased risk of death (Figure 3). 

The primary contributors to the increase in mortality were as follows: duration of hospitalization, kidney injury, cardiovascular comorbidities, and anemia (Hb < 10 g/dL) (Table 4).

The potential confounding factors in the multivariate analysis were as follows: age, sex, duration of diabetes, cardiovascular disease, diminished renal function, anemia, and serum albumin, and the adjustment was made accordingly. The Texas classification was divided into two categories: the first included ulcers from stage A0 to BII, and the second category included ulcers from BIII to DIII. Patients with deep ulcers involving the tendon, joint, or bone had a higher mortality risk than those who had superficial or pre-ulcerative lesions (HR = 1.963, 95% CI: 1.065–3.617) after taking into consideration confounders that could modify the survival time. At the same time, a predictive effect of the associated confounders was observed: age (HR = 1.042, 95% CI: 1.013–1.073), cardiovascular disease (HR = 2.89, 95% CI: 1.540–5.456), anemia (HR = 1.28; 95% CI: 1.281–6.355), and glomerular filtration rate (HR = 2.17; 95% CI: 1.129–4.175) (Table 5). 

Using the same argument, the Wagner–Meggitt classification was divided into two categories: the first category included ulcers with grades 0, 1, and 2, and the second category included those with grades 3, 4, and 5. After inclusion in the multivariate prediction model, patients with bone or joint lesions on admission had an 88.9% higher mortality risk during follow-up (CI: 1.024–3.483) compared with those with superficial ulcers. The presence of cardiovascular comorbidities, chronic kidney disease, anemia, and demographic factors were predictive confounders associated with the risk of mortality (Table 5).

A high grade in the SINBAD Classification System and Score doubled the mortality risk. Furthermore, the presence of a low value of hemoglobin on admission and the association with cardiovascular disease significantly increased the mortality rate during the follow-up period in our sample. 

The SEWSS and WIfI classification system did not have predictive value (HR = 1.823, *p* = 0.067, and HR = 1.755, *p* = 0.077) when other factors influencing survival were taken into account. In this case, survival was influenced by the presence of cardiovascular disease, anemia, and a decreased glomerular filtration rate (Table 5). 

## 4. Discussion

Diabetic foot ulcers represent one of the most incapacitating complications of diabetes and can result in amputations or even death. The treatment and management of diabetic foot ulcers are among the most difficult challenges, not only for patients and their families, but also for medical teams, requiring multimodal and multidisciplinary care [29]. Numerous published studies highlight the relationship between diabetic foot ulcers, cardiovascular events, and increased mortality in patients with diabetes [30]. Diabetic foot is considered a predictor of cardiovascular events and mortality, with possible common pathways.

In patients with diabetes, a close relationship was observed between cardiovascular disease mortality, as well as the severity of diabetic foot ulcers, and a marker of inflammation, but also renal function (elevated cystatin C) [30,31], which reflects the multiorgan and multisystemic lesions caused by neuropathy, vascular damage, and inflammation. The association between DFUs and renal failure confirms the idea that this relationship represents the interconnected evolution of chronic complications, both vascular and neuropathic, and chronic inflammation [30,32].

### 4.1. Survival Rate

In our study, the mortality of patients with diabetic foot ulcers was high (42.2%) and the average survival time was 9 years, with a median of 12 years. The ten-year mortality in patients with DFUs varies in different studies, from 45% [33] to 70% [34]. Cumulative overall survival rates at 1, 5, and 8 years of 95%, 78%, and 66%, respectively, have been reported [35]. In a prospective 10-year study, patients with DFUs or distal amputations had a lower overall survival rate due to cardiovascular events [36].

### 4.2. Cardiovascular Disease as a Predictor of Mortality in DFU Patients

In our study, patients with cardiovascular disease and hypertension had a higher mortality risk. In the multivariate analysis (HR = 1.75, *p* < 0.001 and HR = 1.433, *p* = 0.14, respectively), cardiovascular disease remained an independent predictor of mortality.

In a retrospective cohort study on patients with diabetic foot ulcers, the shortest survival time (999 days) was after major amputation, and age and a history of coronary heart disease were associated with mortality as well [37].

On the other hand, in a study conducted in Spain on 256 patients with DFUs, macrovascular events (stroke, ischemic heart disease, and heart failure) were not related to an increased mortality risk [38]. A meta-analysis concluded that DFUs were associated with an augmented risk of all-cause mortality (RR 1.89, 95% CI 1.60, 2.23) and fatal myocardial infarction (2.22, 95% CI 1.09, 4.53). However, CVD mortality registered a similar proportion in DFU and non-DFU patients [39].

Several ways of associating DFUs with an increase in cardiovascular mortality have been described, but the mechanisms that explain this connection are not fully understood. In diabetic peripheral neuropathy, oxidative stress and chronic inflammation have deleterious effects on the endothelium, in addition to neuronal cells. Chronic inflammation, a prominent characteristic of type 2 diabetes, serves a crucial role in the development of atherosclerosis and vascular complications resulting from endothelial dysfunction [30]. Peripheral neuropathy is involved in the calcification of the arterial media, and the mechanisms include the RANKL/OPG (receptor activator of nuclear factor-κB ligand/osteoprotegerin) pathway, which mediates this process in both the coronary and peripheral arteries [40]. Serum RANKL and tissue RANKL were found to be associated with peripheral arterial calcification, with this association also existing at the carotid level [41,42]. In patients with diabetes, vascular calcification is a strong predictor of lower-limb amputations and cardiovascular mortality as a result of increased arterial stiffness. Vascular calcifications are more frequent in peripheral neuropathy, which is also accompanied by an increase in bone resorption (osteolysis) and changes associated with Charcot’s neuroarthropathy. It has been suggested that the RANK, RANKL, and OPG signaling pathways serve as the link between vascular and bone metabolism [43,44]. Moreover, advanced glycation end-products (AGEs) play an extremely important role in the development of atherosclerotic calcifications in patients with diabetes [45,46]. The major cause of atherosclerosis events in type 2 diabetes is represented by chronic subclinical inflammation. This is related to endothelial dysfunction modulated by cell adhesion molecules, such as E-selectin, leukocyte recruitment and binding dysfunction, macrophage migration alteration, smooth muscle cell proliferation, increased production of angiotensin II, PAI-1, free fatty acids, and AGEs, and increased lipid oxidation, which will result in the further release of proinflammatory cytokines [30,47,48,49,50,51].

### 4.3. Renal Disease as a Predictor of Mortality in Patients with DFUs 

In our study, patients with diabetic nephropathy (HR = 1.36, *p* = 0.12) had a significantly shorter survival time than those without kidney disease. The estimated glomerular filtration rate was an important predictor for the survival of patients with DFUs in the multivariate analysis.

In the study conducted by Aragon-Sanchez et al., an estimated glomerular filtration rate lower than 60 mL/min/1.73 m^2^ increased the mortality risk (HR: 2.2, 95% CI: 1.1–4.2, *p* = 0.01) [35]. Moreover, a low glomerular filtration rate along with major amputation, ischemia, and older age reduced the chance of survival after the diagnosis of diabetic foot infection in a retrospective study that included all adult patients hospitalized between 2010 and 2014 [52].

The mesenchymal cells are progenitors of both renal mesangial cells and smooth muscle vascular cells. Therefore, the pathological processes involved in atherosclerosis will also lead to glomerulosclerosis. Furthermore, these changes lead to a loss of selective permeability of the basal membrane and proteinuria [53].

Cardiometabolic syndrome is characterized by endothelial dysfunction, increased oxidative stress, subclinical proinflammatory status, dyslipidemia, microalbuminuria, and hypertension, all of which contribute to the risk of major cardiovascular events. Glomerulosclerosis and vascular atherosclerosis develop simultaneously in type 2 diabetes, and albuminuria is considered to be predictive for both the progression of diabetic kidney disease and cardiovascular events [30,53]. Furthermore, in type 2 diabetes, aldosterone is crucial in the development of glomerulosclerosis and hypertension through its actions in the brain, heart, blood vessels, and kidneys. Aldosterone acts as a protein kinase C and TNF receptor, and contributes to the development of cardiomyocyte fibrosis and the growth of renal cortical fibroblasts [30]. In addition, it supports decreases in NO-mediated vasodilation and in baroreflex sensitivity, and an increase in sympathetic activity, all of which result in hypertension [30,54].

### 4.4. Ulcer Severity as a Predictor of Mortality in Patients with DFUs

In our sample, 35.1% of patients had superficial lesions, 33.8% had deep ulcers, and 31.1% had wounds that included all layers. Patients with deep ulcers, according to the Texas Staging System, had a higher mortality risk than those with superficial or pre-ulcerative lesions (HR = 1.963, 95% CI: 1.065–3.617). Furthermore, patients with a severe ulcer according to the SINBAD Classification System and Score had a higher mortality risk compared with those who had low- and moderate-grade lesions (HR = 2.333, 95% CI: 1.258–4.326). According to the Wagner–Meggitt classification, patients with bone or joint lesions had an 88.9% higher mortality risk (CI 1.024–3.483). The SEWSS did not have a predictive capacity (HR = 1.823, *p* = 0.067) when the other factors influencing the survival of the patients included in the study were taken into account.

Classification systems for DFUs take into consideration the extension and depth of the lesions and the presence of infection. Early identification and multidisciplinary management of diabetic foot ulcers are crucial in decreasing morbidity and mortality related to limb amputation. In a recent retrospective study conducted by Rubio et al. [55], the SINBAD scoring system was independently associated with mortality. An increasing WIfI stage was correlated with decreasing amputation-free survival at 1 year in a study conducted on a sample of 257 consecutive patients (stage 1: 84%; stage 2: 75%; stage 3: 80%; and stage 4: 69%; *p* = 0.006). However, there was no difference in the overall survival rate between WIfI stages [27].

In a retrospective analysis that included 401 patients between 2010 and 2019, Sen and Demirdal [56] evaluated risk factors for in-hospital mortality in patients with diabetic foot infection. The mortality rate was 3%. In the multivariate analysis, peripheral artery disease (OR: 13.430, 95% CI: 1.129–59.692; *p* = 0.040), thrombocytosis (OR: 1.000, 95% CI: 1.000–1.000; *p* = 0.022), and polymicrobial culture of deep tissues (OR: 7.790, 95% CI: 1.592–38.118; *p* = 0.011) were independent risk factors for mortality [56].

A prospective study conducted by Schofield et al. [57] showed a close relationship between the risk of sepsis, kidney failure, and hind foot ulcer, which was an independent mortality risk factor.

In a retrospective study, Vuorlaakso M et al. [52] reported overall survival rates of 81.2% (95% CI 77.5–84.9%) and 49.7% (95% CI 44.8–54.6%) one and five years after DFU infection, respectively. The presence of ischemia, older age, and increased C-reactive protein diminished survival in patients without major amputation. In patients with major amputation, mortality after an episode of diabetic foot infection remained significantly increased. The results highlighted the major impact of proactive wound management in preventing diabetic foot complications [52].

Despite all efforts to achieve early, complex, and multidisciplinary management of diabetic foot, the rate of wound healing is still diminished and has not increased in the last three decades. In patients with diabetes, their chronic pro-inflammatory status creates an imbalance between the two types of cytokines secreted in wounds, with pro- or anti- inflammatory effects. The primary role in this imbalance is played by the macrophages, as they will secrete different types of cytokines, depending on the phase of the wound-healing process. First, the secretion will consist of several pro-inflammatory cytokines with antimicrobial properties, and afterward, the secretion will switch to anti-inflammatory cytokines, which will promote wound healing. In patients with diabetes, this latter phase is altered, explaining the delayed healing. Thus, the opened wound becomes a gateway for infections that, combined with impaired leukocyte function associated with hyperglycemia, can evolve into sepsis and/or limb amputation [30,45].

### 4.5. Strength and Limitations

We conducted a retrospective study on a sample of patients consecutively hospitalized in the Diabetes, Nutrition, and Metabolic Diseases Clinic at “Sf. Spiridon” Hospital, Iaşi. The design of our study permitted the collection of data from an important number of patients over a significant length of time. According to our knowledge, this is the first comparative analysis of five ulcer severity classification systems in predicting patient mortality using the same sample. These five systems included the oldest ones (Wagner–Meggitt and University of Texas Staging System), but also the newest ones recommended recently by the IWGDF guidelines (SINBAD Classification System and Score, WIfI Classification System). Our findings provide additional real-world evidence for predicting long-term mortality in patients with DFU, which can be added to the previous research in this area. 

However, a prospective design is more appealing, given the possibility to include selected variables to which we did not have access, such as procalcitonin, osteoprotegerin, or RANKL. Also, we could not survey our sample at different points of time to more accurately estimate the glycemic control of our patients. Furthermore, we did not analyze the relationship between DFUs and the cause of death, as we only had access to electronic case histories, which did not register it. Although it would have been important, we did not analyze the relationship with antiplatelet and lipid-lowering medications. Finally, although we conducted our study in a single center, “Sf. Spiridon” Hospital, Iaşi, is a regional center, with a large reach across the north-eastern region of our country, with general surgery and vascular surgery departments, which gave us the opportunity to reduce selection bias.

## 5. Conclusions

In conclusion, the findings of this study indicate that patients presenting with severe diabetic foot ulcers (according to different classification systems) have a significantly greater likelihood of mortality. Even after adjusting for concomitant factors that might influence survival, the observed association remained significant. These findings highlight the significance of recognizing the severity of diabetic foot ulcers at the time of their initial presentation, as they serve as important prognostic indicators. Identifying high-risk patients enables targeted interventions and individualized management strategies to increase their survival chances. Further research is needed to explore additional contributing factors and refine risk assessment models in order to enhance patient outcomes in the management of diabetic foot ulcers.

## Figures and Tables

**Figure 1 healthcare-11-02077-f001:**
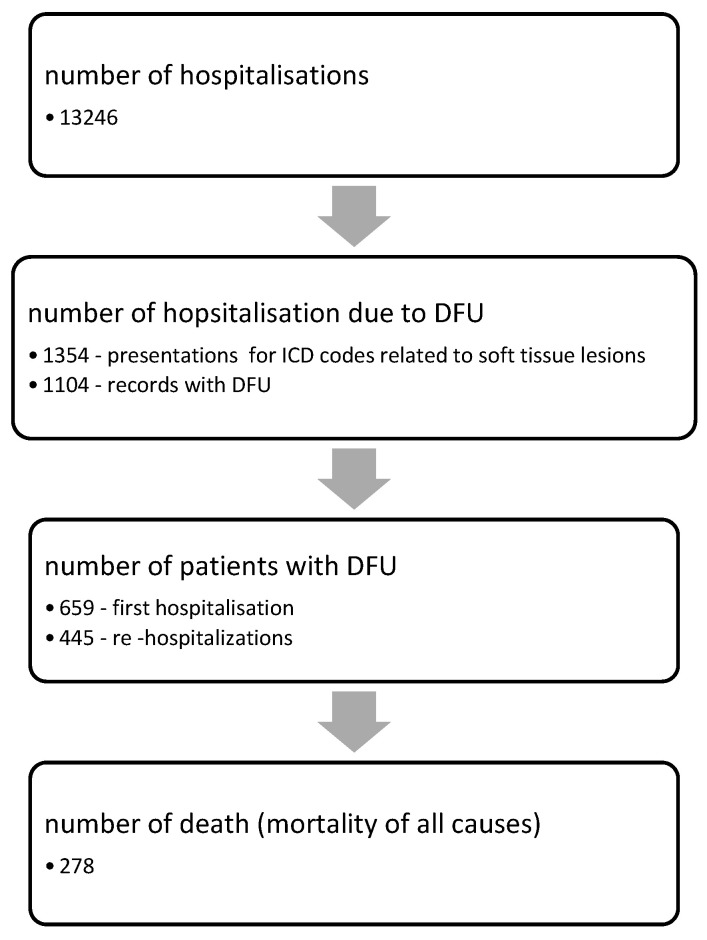
Flow chart of the selection of participants.

**Figure 2 healthcare-11-02077-f002:**
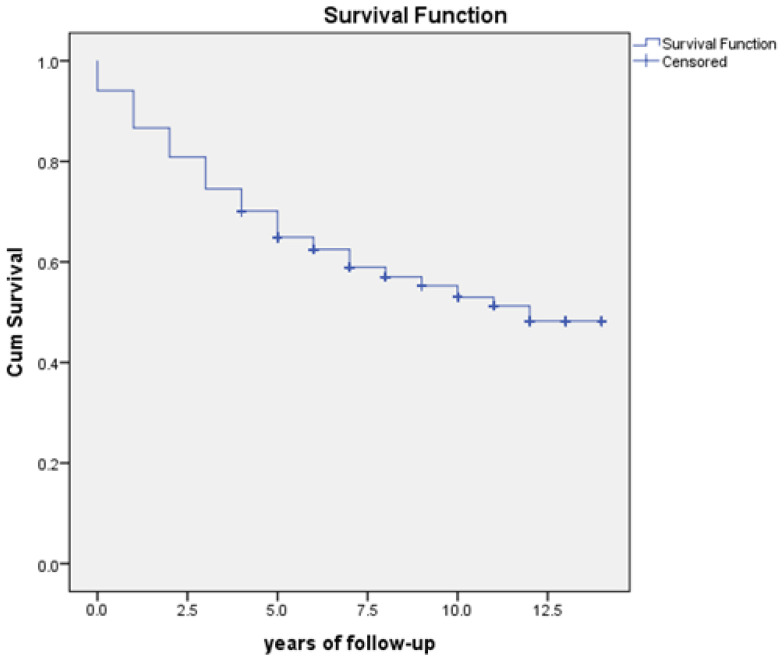
Survival curve of patients with DFUs.

**Figure 3 healthcare-11-02077-f003:**
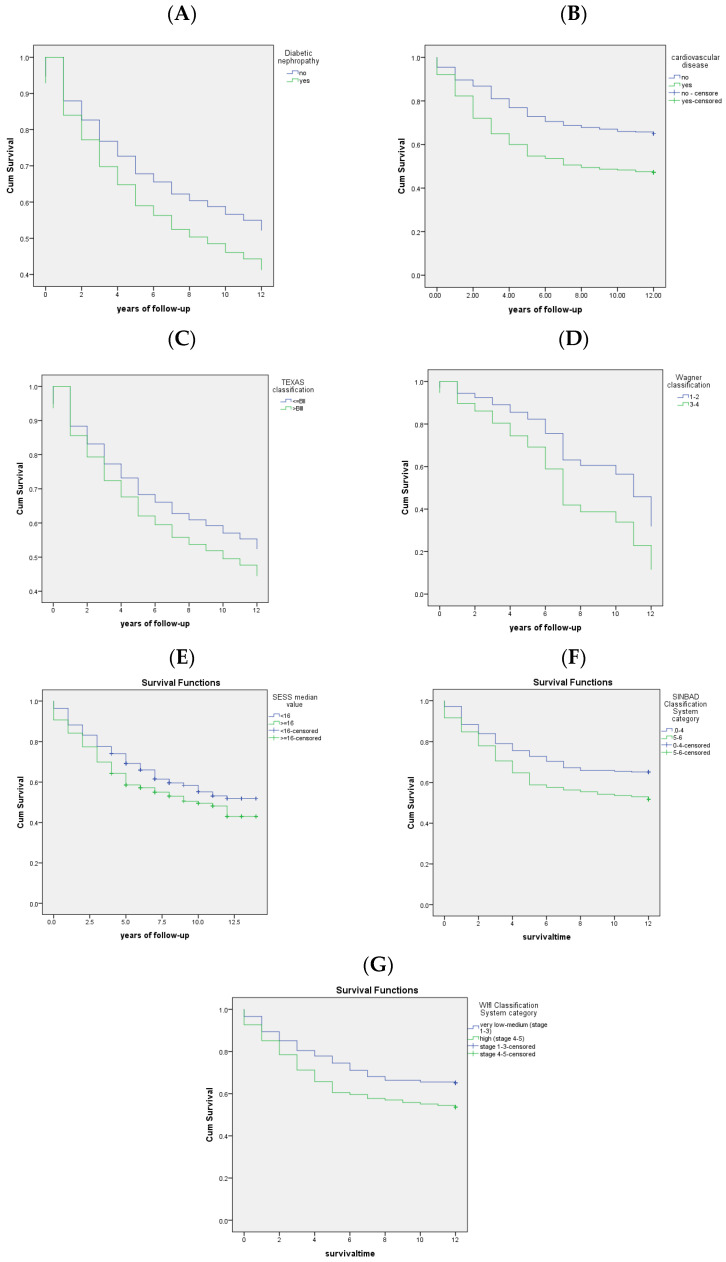
Kaplan–Meier curves for survival ((**A**) diabetic nephropathy; (**B**) cardiovascular disease; (**C**) University of Texas classification; (**D**) Wagner–Meggitt classification; (**E**) Saint Elian Wound Score System; (**F**) SINBAD Classification System and Score; (**G**) WIfI Classification System).

**Table 1 healthcare-11-02077-t001:** Baseline characteristics during the first hospitalization in the Diabetes, Nutrition, and Metabolic Diseases Clinic.

Characteristics	Mean	SD (Std.err.)
Age (years)	61.34	11.062 (0.431)
Years since diagnosis	11.24	8.79 (0.334)
Hospitalization (days)	19.68	13.38(0.522)
Hb (g/dL)	12.45	1.88 (0.07)
Fibrinogen (mg/dL)	478.61	132.89 (5.42)
CRP (mg/dL)	6.72	8.48 (0.46)
Ferritin (µg/mL)	382.20	436.33 (40.86)
HbA1c (%)	9.55	2.19 (0.09)
Albumin (g/dL)	34.85	8.08 (0.68)
Glomerular filtration rate (GFR mL/min/1.73 m^2^)	70.35	26.63 (1.05)
WBC	10,848.39	4897.42 (192.53)
	**N**	**%**
Type of diabetes		
T1DM	80	12.4
T2DM	568	86.19
Other types of DM	11	1.67
Sex (male)	435	66
Area of residence (urban)	339	51.4
Obesity	208	31.6
Diabetic ketoacidosis	98	14.9
DM treatment		
Non-insulin antidiabetic drugs	500	75.9
Insulin alone or in combination with non-insulin therapy	191	29
Diabetic retinopathy	431	65.4
Diabetic nephropathy	210	31.9
Chronic kidney disease	231	35.1
Diabetic neuropathy	619	88.5
Peripheral artery disease	419	63.6
Hypertension	483	73.3
Cardiovascular diseases	394	59.8
Charcot’s foot	21	3.2

**Table 2 healthcare-11-02077-t002:** Allocation of DFUs according to the system of classification.

Classification System	N	%
**Texas classification (stage)**		
A0	6	0.9
AI	74	11.2
AII	10	1.5
AIII	1	0.2
B0	1	0.2
BI	140	21.2
BII	127	19.3
BIII	101	15.3
C0	2	0.3
CI	8	1.2
CII	5	0.8
CIII	23	3.5
D0	3	0.5
DI	17	2.6
DII	40	6.1
DIII	101	15.3
**Wagner–Meggitt classification**		
Grade 0	10	1.5
Grade 1	213	32.4
Grade 2	192	29.2
Grade 3	103	15.7
Grade 4	117	17.8
Grade 5	23	3.5
**SEWSS**		
Mild	35	5.3
Moderate	565	85.7
Severe	59	9.0
**SINBAD Classification System and Score**		
Low grade	287	43.6
Moderate grade	269	40.8
High grade	54	8.2
**WIfI Classification System**		
Stage 1	64	9.7
Stage 2	171	25.9
Stage 3	76	11.5
Stage 4	192	29.1
Stage 5	155	23.5
**Lesions (depth)**		
Superficial	231	35.1
Deep ulcers	223	33.8
Ulcers penetrating all layers	205	31.1

**Table 3 healthcare-11-02077-t003:** Life table of patients included in this study.

Interval Start Time (Years)	Number Entering Interval	Number Exposed to Risk	Cumulative Proportion Surviving at End of Interval	Std. Error of Cumulative Proportion Surviving at End of Interval
5	462	462.000	0.66	0.02
10	393	393.000	0.59	0.02
12	385	194.500	0.57	0.02

**Table 4 healthcare-11-02077-t004:** Independent association with mortality in the univariate analysis of the studied sample.

	*p*.	HR	95.0% CI for HR	
			Lower	Upper
Age	**0.000**	**1.046**	1.034	1.058
Gender	0.073	1.248	0.979	1.591
Area of residence	0.540	0.929	0.734	1.176
Diabetes duration	0.368	1.006	0.993	1.019
**Length of hospitalization**	**0.014**	**1.010**	1.002	1.018
Type 1 DM	0.701	1.262	0.385	4.137
Type 2 DM	0.429	1.584	0.507	4.947
Diabetic retinopathy (DR)	0.475			
DR (nonproliferative)	0.513	0.914	0.700	1.195
DR (proliferative)	0.334	1.207	0.824	1.766
DR (maculopathy)	0.688	1.102	0.686	1.771
**Diabetic nephropathy**	**0.012**	**1.360**	1.069	1.731
**Chronic kidney disease**	**0.004**			
**Stage 1**	**0.035**	**1.407**	1.024	1.935
**Stage2**	**0.001**	**1.698**	1.232	2.341
Stage3	0.777	1.102	0.563	2.157
**Stage 4**	**0.026**	**1.953**	1.085	3.515
Peripheral artery disease	0.204			
Stage 1	0.585	0.728	0.232	2.279
Stage 2	0.153	1.392	0.884	2.192
Stage 3	0.132	1.871	0.828	4.230
Stage 4	0.121	1.234	0.946	1.609
**Cardiovascular diseases**	**0.000**	**1.753**	1.385	2.219
Polyneuropathy	0.729	0.917	0.561	1.498
Charcot’s foot	0.281	0.641	0.285	1.439
**Hypertension**	**0.014**	**1.433**	1.076	1.907
Dyslipidemia	0.320	0.879	0.682	1.133
Obesity	0.303	0.873	0.674	1.131
Diabetic ketoacidosis	0.962	1.008	0.721	1.409
Insulin treatment ± other antidiabetics	0.223	1.181	0.904	1.542
Non-insulin treatment	0.562	1.099	0.799	1.512
**Texas Staging System > IIB**	**0.026**	**1.308**	**1.033**	**1.656**
**SINBAD Classification System and Score—high grade**	**0.001**	**1.520**	**1.182**	**1.954**
**WIfI stage 4–5**	**0.004**	**1.455**	**1.124**	**1.883**
**SEWSS median score**	**0.026**	**1.039**	**1.005**	**1.075**
Wagner–Meggitt classification	0.052	1.269	0.998	1.612
**Lesion depth**	0.015			
**Deep ulcers**	**0.007**	**1.680**	1.151	2.450
**Ulcers intersecting all layers**	**0.023**	**1.567**	1.065	2.306
Cellulitis	0.375			
Cellulitis only in foot	0.430	0.907	0.711	1.157
Cellulitis above the foot	0.203	0.672	0.365	1.239
Fever	0.539	0.870	0.557	1.357
Osteolysis	0.418	1.110	0.862	1.430
Wound culture	0.091			
Negative culture	0.369	0.821	0.533	1.263
Aerobes, Gram-positive	**0.044**	**0.734**	**0.543**	**0.991**
Aerobes, Gram-negative	0.459	1.135	0.811	1.588
Fungi	0.619	1.428	0.351	5.804
**Hb**	**0.015**	**0.925**	**0.868**	**0.985**
**Hb < 10 g/dL**	**0.010**	**1.586**	**1.118**	**2.252**
WBC	0.843	1.000	1.000	1.000
Fibrinogen (mg/dL)	0.325	1.000	0.999	1.000
CRP (mg/dL)	0.490	1.007	0.986	1.029
Glycemia (mg/dL)	0.202	0.999	0.998	1.000
Creatinine (mg/dL)	0.096	1.008	0.999	1.017
**Albumin (g/dL)**	**0.009**	**1.029**	**1.007**	**1.051**
Sideremia (μg/dL)	0.174	0.994	0.986	1.003
**Ferritin (ng/dL)**	**0.002**	**1.001**	**1.000**	**1.001**
LDL cholesterol (mg/dL)	0.841	1.000	0.996	1.004
**GFR (mL/min/1.73 m^2^)**	**0.000**	**0.991**	**0.987**	**0.996**
**GFR < 60 mL/min/1.73 m^2^**	**0.000**	**1.609**	**1.261**	**2.052**
Hemocultures	0.586			
Negative	0.335	1.184	0.840	1.670
Positive	0.756	0.887	0.418	1.883
HbA1c (%)	0.904	1.053	0.457	2.426

Bolded variables represents data with statistical significance (*p* < 0.05).

**Table 5 healthcare-11-02077-t005:** Association with mortality in the multivariate analysis of patients with DFUs.

	*p*.	HR	95.0% CI	
			Inferior	Superior
**Relationship between Texas Staging System and mortality risk (model 1)**				
**Texas (category)**	**0.031**	**1.963**	**1.065**	**3.617**
**Cardiovascular disease**	**0.001**	**2.898**	**1.540**	**5.456**
Hypertension	0.570	0.804	0.379	1.706
**Hb < 10 g/dL**	**0.010**	**2.853**	**1.281**	**6.355**
**GFR < 60 mL/min/1.73 m^2^**	**0.020**	**2.171**	**1.129**	**4.175**
Albumin (g/dL)	0.069	1.034	0.997	1.072
**Age (years)**	**0.004**	**1.042**	**1.013**	**1.073**
Sex (male)	0.408	1.292	0.704	2.370
Diabetes duration (years)	0.228	0.978	0.943	1.014
**Relationship between Wagner–Meggitt classification and mortality risk (model 2)**				
**Wagner–Meggitt (category)**	**0.042**	**1.889**	**1.024**	**3.483**
**Cardiovascular disease**	**0.001**	**2.829**	**1.489**	**5.376**
Hypertension	0.589	0.812	0.382	1.727
**Hb < 10 g/dL**	**0.029**	**2.508**	**1.098**	**5.733**
**GFR < 60 mL/min/1.73 m^2^**	**0.006**	**2.470**	**1.297**	**4.705**
Albumin (g/dL)	0.091	1.032	0.995	1.070
**Age (years)**	**0.006**	**1.041**	**1.011**	**1.071**
Sex (male)	0.341	1.347	0.730	2.485
Diabetes duration (years)	0.247	0.979	0.943	1.015
**Relationship between SEWSS and mortality risk (model 3)**				
**SEWSS (median score)**	0.067	1.823	0.959	3.466
**Cardiovascular disease**	**0.003**	**2.717**	**1.407**	**5.247**
Hypertension	0.678	0.852	0.399	1.816
**Hb < 10 g/dL**	**0.016**	**2.722**	**1.208**	**6.135**
**GFR < 60 mL/min/1.73 m^2^**	**0.006**	**2.454**	**1.297**	**4.645**
Albumin (g/dL)	0.068	1.035	0.997	1.074
**Age (years)**	**0.002**	**1.048**	**1.018**	**1.079**
Sex (male)	0.264	1.408	0.772	2.569
Diabetes duration (years)	0.119	0.969	0.932	1.008
**Relationship between SINBAD and mortality risk (model 4)**				
**SINBAD (score 5–6)**	**0.007**	**2.333**	**1.258**	**4.326**
**Cardiovascular disease**	**0.002**	**2.967**	**1.472**	**5.975**
Hypertension	0.161	0.580	0.271	1.243
**Hb < 10 g/dL**	**0.001**	**5.939**	**2.605**	**13.54**
GFR < 60 mL/min/1.73 m^2^	0.070	1.916	0.949	3.869
Albumin (g/dL)	0.065	1.049	0.997	1.103
Age (years)	0.102	1.026	0.995	1.058
Sex (male)	0.319	1.374	0.735	2.568
**Diabetes duration (years)**	**0.030**	**0.953**	**0.913**	**0.995**
**Relationship between WIfI and mortality risk (model 5)**				
WIfI (stage 4–5)	0.077	1.755	0.940	3.277
**Cardiovascular disease**	**0.001**	**2.956**	**1.569**	**5.568**
Hypertension	0.370	0.708	0.333	1.506
**Hb < 10 g/dL**	**0.002**	**3.479**	**1.589**	**7.613**
GFR < 60 mL/min/1.73 m^2^	0.079	1.037	0.937	3.274
Albumin (g/dL)	0.058	1.035	0.999	1.076
**Age (years)**	**0.024**	**1.033**	**1.004**	**1.063**
Sex (male)	0.195	1.494	0.814	2.741
Diabetes duration (years)	0.120	0.970	0.933	1.008

Bolded variables represents data with statistical significance (*p* < 0.05).

## Data Availability

Data used in this study will be available from the corresponding authors upon request.

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
