# Peer review of "Evaluating Classification Systems of Diabetic Foot Ulcer Severity: A 12-Year Retrospective Study on Factors Impacting Survival"

_healthcare, 2023, doi:10.3390/healthcare11142077_

Round 1

Reviewer 1 Report

well written however it is a bit complicated

in discussion

the aim of article should be written in first paragraph

sectıon  285  number cystatın c, lipotoxicity should be explained or deleted what is the relationship of this materials about classification types and also if it is needed should be explained and shown in result section and tables

also in number 310 rankl/opg and other pathways are not sufficient it is only pathophysiologic way should not be explained there

I couldn't find where the ratio between deep ulcers and death was mentioned in the article.

but in the conclusion part, the author has finished the conclusion part especially with this subject. 

and also  in the abstract section, it is written as ''Length of hospitalization, diabetic nephropathy, chronic kidney disease, glomerular filtration rate, cardiovascular disease, hypertension, anemia, and DFU severity, were the most significant contributors to the increase in mortality''. not mentioned about deep ulcers of foot 

mınor edıtıng should be replaced

Author Response

We attached a file with our response.

Reviewer 2 Report

I have attached my considerations.

No particulary comments

Author Response

We attached a file with the response.
